# Transformer Depth Reduction via Low-Pass Filtering

## Abstract

We introduce *Time-Variable Low-Pass* (TVLP) filtering of attention projections and apply it to a GPT-3 small-style Transformer. We implement TVLP with custom sequential Metal kernels on Apple M4 Pro (MPS) and demonstrate that a depth-11 model with TVLP applied across all layers exhibits lower bits-per-byte (BPB) than the depth-11 baseline and nearly matches the depth-12 baseline at equal training steps throughout the Chinchilla regime. Measured at equal FLOPs, the TVLP-enhanced depth-11 model shows lower BPB than both baselines, while at equal wall-clock time it achieves lower-or-equal BPB throughout the training process. We also implement TVLP with custom warp-scan CUDA kernels and confirm the wall-clock advantage at training speeds measured on Nvidia L40S. Notably, the parameter count, inference cache size, and FLOPs per token added by TVLP across all 11 layers represent only 2.42% of the number of parameters, 1.61% of the size of the KV-cache, and 2.11% of the FLOPs per token in a single standard Transformer block, respectively.

## 1 Introduction

Transformer models Vaswani et al. (2017) have become the dominant architecture for large-scale sequence modeling, achieving strong performance through deep stacks of attention and feed-forward layers. However, increasing depth incurs substantial training cost and memory usage at inference time, motivating efforts to improve parameter efficiency without sacrificing effective model capacity Dehghani et al. (2019); Lan et al. (2020). Several lines of work have explored augmentations or alternatives to standard attention. These include architectural modifications to improve efficiency Tay et al. (2022); Wu et al. (2020); Beltagy et al. (2020), as well as approaches that introduce implicit recurrence, memory, or state Dai et al. (2019); Gu & Dao (2024); Su et al. (2024). A common theme is that temporal structure, in particular the separation of short- and long-range dependencies, can be exploited to reduce reliance on deep stacking of Transformer blocks.

In this work, we propose *Time-Variable Low-Pass* (TVLP) filtering of attention projections, a simple mechanism that applies a causal, input-dependent exponential moving average to queries, keys, and values prior to positional encoding and attention. At each timestep, a gating coefficient which controls the degree of temporal smoothing is computed from the input, enabling the model to adaptively filter high-frequency components. TVLP can be applied independently to all attention layers with negligible parameter overhead.

We empirically evaluate TVLP within the NanoChat framework Karpathy (2025), a modern pedagogical harness designed for training compute-optimal transformers. To isolate and assess TVLP's depth substitution capabilities, we maintain a constant number of heads and a fixed head dimension across all experiments. We use a head dimension slightly larger than the standard NanoChat recommendation to preclude the possibility of model width-starvation. Tested on Apple M4 Pro (MPS) with custom Metal kernels, a depth-11 configuration with TVLP applied to all layers nearly matches the performance of the depth-12 baseline at equal training steps. This is achieved with only a modest parameter and inference cache increase relative to the depth-11 baseline, and substantially fewer parameters and smaller inference cache than the deeper baseline. The TVLP model consistently outperforms both baselines at equal training FLOPs. At equal wall-clock time, it maintains a slight advantage during the first two-thirds of the training process, and sustains a lower or equal evaluation BPB thereafter. We show that these advantages continue to hold when adjusted for the training speeds measured on Nvidia L40S hardware with our custom CUDA kernels.

These experiments confirm that the TVLP advantage at compute-theoretical FLOPs translates to a practical wall-clock advantage on multiple hardware architectures.

We further observe that the first layer (layer-0) segregates attention heads into distinct memory regimes, including long-term memory, whereas deeper layers only exhibit short- and medium-term memory in a strongly input-dependent manner. These findings indicate that TVLP introduces a structured temporal inductive bias that complements attention and partially substitutes for physical depth, reminiscent of findings in recurrent and state-augmented Transformer variants Dai et al. (2019); Su et al. (2024).

**Contributions**

- We introduce TVLP, an input-dependent temporal smoothing applied to attention projections.

- We show that TVLP enables reduction of physical depth while maintaining effective depth, with a depth-11 model nearly matching a depth-12 baseline at equal training steps.

- We demonstrate improved training efficiency both at equal training FLOPs and at equal wall-clock time on Apple M4 Pro (MPS), a finding which extrapolates to the training speeds measured on Nvidia L40S (CUDA).

- Finally, we provide empirical analysis of head specialization into distinct memory regimes, including long-term memory in the first layer, and input-dependent short- and medium-term memory in subsequent layers.

## 2 TVLP Mechanism

For a generic input sequence $\mathbf{x}_t \in \mathbb{R}^d$, the filtered output $\mathbf{h}_t \in \mathbb{R}^d$ is defined recursively as a causal exponential moving average:

$$\mathbf{h}_t = (1 - \alpha_t)\,\mathbf{h}_{t-1} + \alpha_t\,\mathbf{x}_t, \tag{1}$$

where $\alpha_t \in (0, 1)$ and $\mathbf{h}_{-1} \in \mathbb{R}^d$ is the initial state.

**Time-variable gating**  The time-varying gating coefficient $\alpha_t$ is input-dependent and is computed as

$$\alpha_t = \sigma\big(\mathbf{w}_\alpha^\top \mathbf{x}_t + b_\alpha\big), \tag{2}$$

where $\mathbf{w}_\alpha \in \mathbb{R}^d$, $b_\alpha \in \mathbb{R}$, and $\sigma(\cdot)$ is the logistic sigmoid.

**Application to attention projections**  Given standard linear projections

$$\mathbf{q}_t = \mathbf{W}_q \mathbf{x}_t, \quad \mathbf{k}_t = \mathbf{W}_k \mathbf{x}_t, \quad \mathbf{v}_t = \mathbf{W}_v \mathbf{x}_t, \tag{3}$$

we apply the filter independently to each attention projection of each attention head.

**Initialization**  Let

$$z_t^{(*)} = \mathbf{w}_\alpha^{(*)\top} \mathbf{x}_t + b_\alpha^{(*)} \tag{4}$$

denote the pre-sigmoid logits for $* \in \{q, k, v\}$. The weights $\mathbf{w}_\alpha^{(*)} \in \mathbb{R}^d$ are initialized from a zero-centered uniform distribution with scale matched to the other projection matrices in the model. The biases $b_\alpha^{(*)} \in \mathbb{R}$ are initialized as $b_\alpha^{(*)} = 0$. The initial states $\tilde{\mathbf{q}}_{-1}, \tilde{\mathbf{k}}_{-1}, \tilde{\mathbf{v}}_{-1} \in \mathbb{R}^d$ are learned parameters, initialized from a zero-centered distribution with the same scale as the other projection weights in the model. During auto-regressive inference, these states are initialized from the cached values of the previous time step.

# 3 Methods

We started from NanoChat git commit ae0bf52 (2026-01-05), and also included the combined AdamW-Muon optimizer from the later git commit 41bb2ea (2026-01-29). The compute-optimal number of training steps displayed in the figures below was computed based on git commit ccf4b7f (2026-01-07), which estimates the optimal ratio of tokens to parameters for this version of NanoChat to be approximately 8.

We combined the separate query (Q), key (K), and value (V) linear nodes into a single linear node, which slightly improved the baseline training speed. The training sequence length was fixed to $T = 1024$ and the training batch size was set to $B = 4$ for all the experiments. The number of heads was set to $H = 8$ and the head dimension to $D = 128$, resulting in a total model width of 1024. This is slightly larger than the optimal width of 768 which NanoChat computes for a depth of 12.

With these settings, the resulting baseline performance and parameter counts correspond roughly to those of small-style GPT-3 models. We set the number of training steps to 10,880 in all cases, which corresponds to the Chinchilla training regime Hoffmann et al. (2022) for the depth-12 baseline configuration. TVLP was applied before the rotary embeddings, which are already present in the NanoChat baseline. The KV cache was updated to also store the Q, K, and V from the latest timestep before positional encoding. The TVLP parameters followed the standard optimizer split: the weights $w_\alpha$ were optimized with Muon as matrix-valued parameters, while the biases $b_\alpha$ and the learned initial states (e.g., $\tilde{q}_{-1}, \tilde{k}_{-1}, \tilde{v}_{-1}$) were optimized with AdamW as vector-valued parameters.

The training FLOPs per token for the baseline models were estimated as per Chowdhery et al. (2023), while for TVLP we added the computation of the temporal filter. Specifically, the dense parameter count associated with the gating logic is modeled at 6 FLOPs per parameter, while the causal exponential moving average imposes an additional compute footprint of 7 FLOPs per active element when including the combined operations for the forward and backward passes (3 operations for the forward integration and 4 operations for the gradient calculations).

## 3.1 Custom PyTorch operators

**Apple M4 Pro (MPS)**  We implemented TVLP using custom Metal kernels implementing an $O(T)$ sequential algorithm, which was found to be sufficient for our target sequence length. The forward and backward kernels used identical launch grid dimensions of $(B, 3H) = (4, 24)$ with a thread group size of $D = 128$.

**Nvidia L40S (CUDA)**  On Nvidia L40S, the sequential strategy was found to provide insufficient parallelism; therefore, we implemented TVLP as CUDA kernels that additionally parallelize over the time dimension using warp scans provided by the Nvidia CCCL library. **Forward:** The forward kernel used a grid size of $(B, 3H, R_f)$ and a block size of $(L_f, P_f)$, where $L_f = 8$ is the logical warp size and $P_f = 8$ is the number of parallel logical warps within the thread block. Each thread processed $I_f = 4$ elements, while $R_f = 4$ according to $D = R_f P_f I_f$. **Backward:** The backward kernel used a grid size of $(B, 3H, R_b)$ and a block size of $(L_b, P_b)$, where $L_b = 32$ is the logical warp size and $P_b = 2$ is the number of parallel logical warps within the thread block. Each thread processed $I_b = 8$ elements, while $R_b = 8$ according to $D = R_b P_b I_b$. **Matrix multiplication:** Because `torch.nn.Linear` cannot produce time-contiguous tensors, which the warp scan operates on most efficiently, we implemented a custom Torch operator for matrix multiplication. The operator invokes `cublasGemmStridedBatchedEx` once during the forward pass and `cublasGemmEx` twice per batch element during the backward pass, which we found to be sufficiently fast for $B = 4$.

On both MPS and CUDA, the inputs and outputs of the custom TVLP kernels were of type `bfloat16`, while internal accumulations were performed using `float32`.

The model source code is accessible at `https://anonymous.4open.science/r/nanochat-fork-anon`.

### 3.2 Time measurements

**Apple M4 Pro (MPS)** As training speed on MPS we used the number of tokens processed per second in the second step of a training run, a metric which we found to be of sufficiently low variance.

**Nvidia L40S (CUDA)** The training speed on Nvidia L40S displayed high variance, so we measured it by launching a short training session 8 times, each time training for 4 steps. For each training run we took the maximum number of tokens processed per second among the 4 steps, and then computed the average of the maxima over the 8 runs.

### 3.3 Effective memory

**Dynamic effective memory length** The effective memory of TVLP is intrinsically sequence-dependent. To quantify the temporal depth of the filter, we define the *dynamic effective memory length*, $\tau_t$, as the expected temporal support length of the input information retained in the current state $\mathbf{h}_t$. Given the causal recurrence established in Equation 1, the contribution of a past input $\mathbf{x}_{t-k}$ to the current state $\mathbf{h}_t$ is governed by the weight

$$w_{t,t-k} = \alpha_{t-k} \prod_{j=0}^{k-1}(1 - \alpha_{t-j}),$$

for $0 \le k \le t$. For finite sequences the total contribution of historical inputs does not generally sum to 1 due to the residual contribution of the initial state $\mathbf{h}_{-1}$. We compute therefore the dynamic effective memory length as

$$\tau_t = \sum_{k=0}^{t}(k+1)\, w_{t,t-k} \Big/ \sum_{k=0}^{t} w_{t,t-k}. \tag{5}$$

Periods of strong low-pass filtering (small $\alpha_t$) increase the effective memory length by preserving past states, whereas large $\alpha_t$ values rapidly overwrite the state with current information.

**99% effective memory span** In addition to the expectation-based memory length of Equation (5), we also compute the minimum temporal horizon required to capture a fixed fraction of the retained mass. Specifically, we define the 99% *effective memory span, $s_{0.99}(t)$* as:

$$s_{0.99}(t) = 1 + \min\left\{ 0 \le k \le t \;\middle|\; \sum_{i=0}^{k} w_{t,t-i} \ge 0.99 \sum_{i=0}^{t} w_{t,t-i} \right\}. \tag{6}$$

This quantity measures the shortest backward temporal span length, in tokens, that contains 99% of the total historical contribution to the current state.

## 4 Experiments

**Projection set** We first evaluated TVLP by enabling filtering across all eight possible subsets of the $\{Q, K, V\}$ projections in a depth-1 model trained in the Chinchilla regime. We found that at the final step, the full set $\{Q, K, V\}$ yielded the lowest evaluation bits-per-byte (BPB), and the only configuration which resulted in model divergence was when TVLP was enabled for $V$ only. This finding was also confirmed in a subsequent test with depth 11.

**Model summary** We trained four different model configurations: two baselines with depth 11 and 12, respectively, and two depth-11 configurations with TVLP applied to the full set $\{Q, K, V\}$: once only for the first layer (layer-0), and once for all the 11 layers. We trained all four model variants on Apple M4 Pro for the number of steps dictated by the Chinchilla regime for the model variant with the highest number of parameters, namely the depth-12 baseline. On Nvidia L40S we only trained for four steps to compute the training speed as described in Section 3.2. All four model variants are summarized in Table 1.

Table 1: Summary of the trained model variants, displaying depth, number of parameters, training speed and inference cache size. The latter comprises the KV cache and, for the TVLP-enabled layers, the cached Q, K and V from the latest timestamp. In each numerical column the best values are shown in bold, while the least preferable values are shown in italics.

| Variant | Depth | Parameters | FLOPs/tok | Cache (MiB) | Apple M4 Pro (tok/s) | Nvidia L40S (tok/s) |
|---|---|---|---|---|---|---|
| Baseline (11L) | 11 | **272,629,760** | **$1.371{,}5\times10^9$** | **44.000** | **4,490** | **96,942** |
| Baseline (12L) | 12 | *285,212,672* | *$1.459{,}6\times10^9$* | *48.000* | *4,201* | *91,423* |
| TVLP (all layers) | 11 | 272,934,152 | $1.373{,}4\times10^9$ | 44.064 | 4,348 | 94,284 |
| TVLP (layer-0) | 11 | 272,657,432 | $1.371{,}7\times10^9$ | 44.006 | 4,461 | 96,759 |

## 4.1 Results

**Equal training steps**  The training loss curves of the four model variants show similar progressions, as seen in Figure A.1, while the evaluation BPB curves show a clear separation, as illustrated in Figure 1: the lower-BPB curves belong to the depth-12 baseline and TVLP applied to all layers, while the higher-BPB curves are the depth-11 baseline and TVLP applied only to layer-0. The CORE metric, measured at several training checkpoints, spans a similar range across all four model variants but exhibits substantial variance, as shown in Figure A.2. We therefore use evaluation BPB as the primary metric for model comparison.

**Equal FLOPs**  Because the depth-11 model with TVLP requires only marginally more FLOPs per token than the depth-11 baseline while achieving an evaluation BPB nearly matching the depth-12 baseline, it consistently outperforms both baselines when evaluated at equal FLOPs (see Figure 2).

**Equal wall-clock time on Apple M4 Pro (MPS)**  Using the measured training speed of Table 1, we plot in Figure 3 the evaluation BPB as a function of wall-clock time on Apple M4 Pro (MPS). During the first two-thirds of the total training time, including at the estimated compute-optimal step, the model with TVLP applied to all layers achieves the lowest evaluation BPB across all four variants. In the final third of the training time, the depth-11 baseline and the model with TVLP applied only to layer-0 display similarly low evaluation BPB; however, their loss descent decelerates toward the end, likely due to their more limited effective capacity.

**Equal wall-clock time on Nvidia L40S (CUDA)**  By adjusting the wall-clock time of Figure 3 to the training speed corresponding to Nvidia L40S (CUDA) in Table 1, we obtain a qualitatively similar result, as seen in Figure A.3. Indeed, while the absolute training speeds on Nvidia L40S are higher, with our implementation the training speed ratios between the four different models remain similar to those on Apple M4 Pro (MPS).

# 5 Structural insights

## 5.1 Effective memory statistics

We computed the *dynamic effective memory length* of the depth-11 model with TVLP applied to all layers for a large sample from the evaluation data set using Equation (5), and report in Table 2 the *mean* values thereof at time step 1023. It can be observed that layer-0 has one head with long-term memory for K (457 timesteps) and V (489 timesteps) on average, and one head with mean medium-term memory for K (29.7 timesteps) and short-term memory for Q (9.6 timesteps) and V (9.8 timesteps) on average. Intermediate layers have mostly one head each with mean short-term memory for Q, while the last 3 layers each have a head which remembers on average the previous 1 or 2 V values.

Since the effective memory is input-dependent, we also investigated how much memory each layer head uses not only on average, but also at the 99th percentile. We computed using Equation (6) the *99% effective*

Figure 1: Evaluation BPB of the four model variants of Table 1 trained on Apple M4 Pro (MPS). The points of the depth-11 model with TVLP applied to all layers are connected through a continuous red line, while those of the depth-11 baseline are connected through a dashed black line. The two lines clearly separate, with the former being shifted significantly lower than the latter and almost as low as the depth-12 baseline. The model with TVLP applied only to layer-0 shows performance very similar to the depth-11 baseline. The doted vertical line indicates the approximate optimal-compute number of training steps corresponding to the depth-12 baseline.

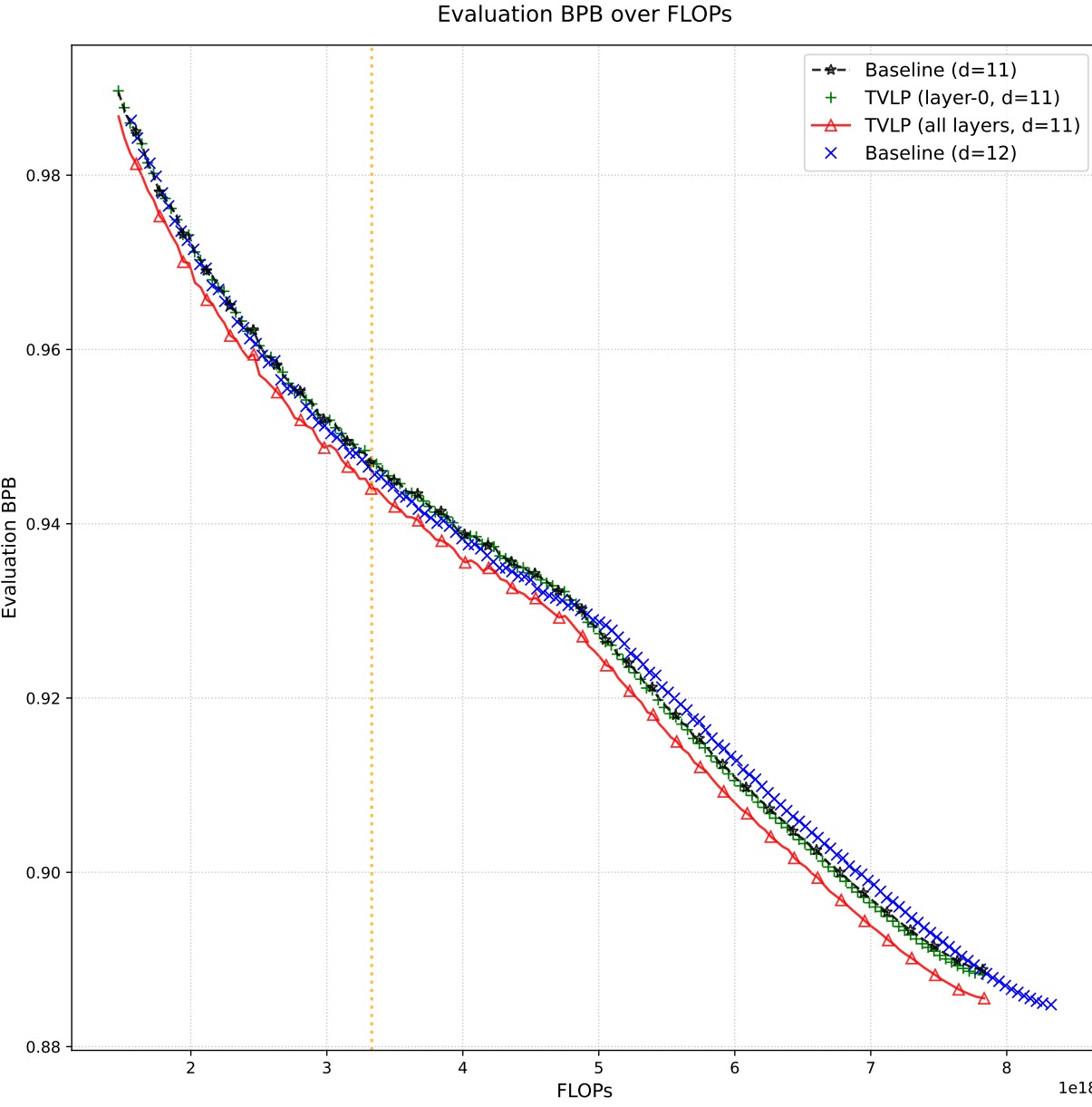

Figure 2: Evaluation BPB of all four model variants at equal FLOPs. This figure was obtained by multiplying the number of training steps on the X axis of Figure 1 with the number of tokens per training step - the constant number 524,288 - and with the number of FLOPs per token from Table 1. Due to its evaluation-BPB advantage over the depth-11 baseline and its FLOP-speed advantage over the depth-12 baseline, the model variant with TVLP applied to all 11 layers consistently outperforms the other model variants at equal FLOPs. The line styles are like in Figure 1.

Figure 3: Evaluation BPB of all four model variants at equal wall-clock time on Apple M4 Pro (MPS). The depth-11 model with TVLP applied to all layers maintains a lower or equal evaluation BPB throughout training, demonstrating a slight advantage during the first two-thirds of the training time, including at the estimated compute-optimal step indicated by a vertical dotted line. Towards the end it plateaus later than its same-depth counterparts, suggesting that its effective capacity is more akin to that of the deeper baseline. The line styles are like in Figure 1.

Table 2: Mean dynamic effective memory length at time step 1023 represented as a vector $\mathbf{m} = (m_Q, m_K, m_V)$ for the depth-11 model with TVLP applied to all layers. The 8 heads are sorted within each layer row by increasing vector norm $\|\mathbf{m}\|_2$. Layer-wise extrema are bold (max) / italic (min), while global extrema across all layers and heads for each of Q, K and V are underlined.

| Layer | Heads sorted by $\|\mathbf{m}\|_2$ | | | | | | | | | | | | | | | | | | | | | | | |
|---|---|---|---|---|---|---|---|---|---|---|---|---|---|---|---|---|---|---|---|---|---|---|---|---|
| | $m_Q$ | $m_K$ | $m_V$ | $m_Q$ | $m_K$ | $m_V$ | $m_Q$ | $m_K$ | $m_V$ | $m_Q$ | $m_K$ | $m_V$ | $m_Q$ | $m_K$ | $m_V$ | $m_Q$ | $m_K$ | $m_V$ | $m_Q$ | $m_K$ | $m_V$ | $m_Q$ | $m_K$ | $m_V$ |
| 0 | 1.0 | 1.2 | *1.0* | *1.0* | *1.0* | 1.3 | 1.1 | 1.2 | 1.1 | 1.2 | 1.1 | 1.1 | 1.2 | 1.2 | 1.3 | 1.5 | 4.0 | 1.1 | **9.6** | 29.7 | 9.8 | 1.3 | **457.0** | **489.0** |
| 1 | *1.0* | 1.0 | *1.0* | 1.1 | *1.0* | 1.0 | 1.3 | 1.0 | 1.1 | 1.3 | 1.1 | 1.1 | 1.2 | **1.3** | 1.0 | 1.6 | 1.0 | 1.1 | 1.5 | 1.1 | **1.1** | **1.6** | 1.1 | 1.1 |
| 2 | 1.2 | 1.1 | *1.0* | *1.1* | 1.1 | 1.0 | 1.2 | 1.1 | 1.0 | 1.3 | **1.2** | 1.0 | 1.3 | 1.1 | 1.1 | 1.4 | 1.1 | **1.2** | 1.8 | 1.2 | 1.0 | **5.4** | *1.0* | 1.0 |
| 3 | 1.4 | *1.0* | 1.0 | 1.4 | 1.1 | 1.1 | *1.2* | **1.5** | 1.0 | 1.2 | 1.5 | 1.0 | 1.7 | 1.0 | *1.0* | 1.7 | 1.2 | 1.2 | 2.0 | 1.1 | 1.2 | **2.2** | 1.1 | **1.3** |
| 4 | *1.2* | *1.0* | *1.0* | 1.5 | 1.1 | 1.1 | 1.5 | 1.1 | 1.1 | 1.4 | **1.4** | 1.1 | 1.8 | 1.1 | 1.1 | 1.6 | 1.2 | 1.3 | 1.7 | 1.1 | 1.3 | **4.8** | 1.1 | **1.7** |
| 5 | 1.1 | 1.0 | 1.0 | *1.1* | 1.4 | *1.0* | 1.4 | 1.1 | 1.2 | 1.5 | 1.1 | 1.0 | 1.9 | 1.2 | **1.3** | 1.8 | **1.7** | 1.1 | 3.2 | 1.1 | 1.2 | **5.3** | *1.0* | 1.0 |
| 6 | *1.1* | 1.0 | 1.0 | 1.2 | 1.0 | 1.1 | 1.2 | 1.1 | 1.2 | 1.2 | 1.0 | **1.3** | 1.4 | 1.0 | 1.1 | 1.1 | 1.3 | 1.1 | 1.2 | **1.3** | 1.1 | **1.7** | *1.0* | *1.0* |
| 7 | *1.0* | 1.0 | 1.0 | 1.1 | 1.0 | 1.0 | 1.2 | 1.0 | 1.2 | 1.5 | *1.0* | *1.0* | 1.1 | 1.3 | 1.6 | 1.5 | 1.0 | 1.5 | 1.4 | **1.3** | **2.2** | **3.2** | 1.1 | 1.0 |
| 8 | 1.0 | 1.0 | 1.0 | 1.1 | 1.0 | 1.1 | 1.0 | 1.0 | 1.3 | 1.1 | 1.1 | 1.2 | 1.1 | 1.0 | 1.2 | **1.4** | *1.0* | *1.0* | 1.1 | 1.1 | 1.3 | *1.0* | **1.4** | **2.8** |
| 9 | 1.1 | *1.0* | 1.0 | 1.1 | 1.0 | 1.0 | 1.0 | 1.0 | 1.1 | 1.1 | 1.0 | 1.2 | 1.4 | 1.0 | *1.0* | **1.4** | 1.0 | 1.2 | *1.0* | 1.1 | 1.8 | 1.1 | **1.4** | **2.4** |
| 10 | 1.0 | 1.0 | *1.0* | 1.0 | *1.0* | 1.1 | *1.0* | 1.0 | 1.2 | **1.0** | 1.0 | 1.2 | 1.0 | 1.1 | 1.3 | 1.0 | 1.1 | 1.6 | 1.0 | **1.2** | 1.9 | 1.0 | 1.0 | **2.6** |

*memory span*, and show in Table 3 the *99th percentile* thereof. Layer-0 utilizes the largest effective memories, with one head occasionally retaining practically the entire sequence in K and V, and another head also retaining significant portions of Q, K, and V. Intermediate layers sporadically retain significant portions of Q, while the last 3 layers show a preference for retaining V values.

Table 3: 99th percentile of the 99% effective memory span at time step 1023, represented as a vector $\mathbf{m} = (m_Q, m_K, m_V)$, for the model with TVLP applied to all layers. Heads are sorted within each layer row by increasing vector norm $\|\mathbf{m}\|_2$. Layer-wise extrema are bold (max) / italic (min), while global extrema across all layers and heads for each of Q, K and V are underlined.

| Layer | Heads sorted by $\|\mathbf{m}\|_2$ | | | | | | | | | | | | | | | | | | | | | | | |
|---|---|---|---|---|---|---|---|---|---|---|---|---|---|---|---|---|---|---|---|---|---|---|---|---|
| | $m_Q$ | $m_K$ | $m_V$ | $m_Q$ | $m_K$ | $m_V$ | $m_Q$ | $m_K$ | $m_V$ | $m_Q$ | $m_K$ | $m_V$ | $m_Q$ | $m_K$ | $m_V$ | $m_Q$ | $m_K$ | $m_V$ | $m_Q$ | $m_K$ | $m_V$ | $m_Q$ | $m_K$ | $m_V$ |
| 0 | *2* | *4* | *2* | *2* | *2* | 5 | 5 | 4 | 3 | 5 | 4 | 4 | 5 | 4 | 5 | 7 | 21 | 4 | **56** | 163 | 51 | 5 | **1024** | **1017** |
| 1 | *2* | *2* | *2* | 3 | *2* | *2* | 5 | 3 | 3 | 6 | 3 | 3 | 5 | **6** | *2* | 7 | 3 | 3 | 8 | 3 | **4** | **9** | 4 | 3 |
| 2 | *4* | *3* | *1* | *4* | *3* | 2 | *4* | **4** | 2 | 5 | 4 | 2 | 6 | *3* | **4** | 6 | *3* | 4 | 14 | 4 | 3 | **49** | *3* | 2 |
| 3 | 6 | *3* | 3 | 6 | *3* | 3 | *4* | **6** | 3 | 5 | 6 | 3 | 9 | 5 | 5 | 11 | *3* | 2 | 11 | *3* | 4 | **12** | 4 | **5** |
| 4 | *4* | *2* | *2* | 7 | 3 | 3 | 7 | 3 | 4 | 9 | 3 | 3 | 10 | 4 | 5 | 10 | 4 | 6 | 8 | **9** | 4 | **33** | 4 | **10** |
| 5 | 5 | 3 | 2 | *4* | 5 | *1* | 7 | 3 | 3 | 6 | 4 | 5 | 11 | 5 | 5 | 10 | **10** | 4 | 21 | 4 | **5** | **54** | *2* | 3 |
| 6 | *4* | *2* | *2* | 5 | *2* | 5 | 6 | *2* | 4 | 5 | 3 | 5 | 5 | 5 | 4 | 5 | 4 | 5 | 5 | **6** | 3 | **10** | *2* | *2* |
| 7 | *3* | 3 | 2 | *3* | 3 | 3 | 4 | 3 | 6 | 9 | *1* | *1* | 4 | 5 | 7 | 9 | 3 | 9 | 7 | **7** | 15 | 22 | 4 | 3 |
| 8 | *3* | *1* | *2* | *3* | 3 | 4 | *3* | 2 | 6 | 4 | 4 | 5 | 4 | 3 | 6 | **8** | *1* | 2 | 5 | 3 | 6 | *3* | **8** | 17 |
| 9 | 3 | *1* | 2 | 3 | 2 | 3 | 3 | 3 | 4 | 3 | 2 | 5 | **7** | *1* | *1* | **7** | 3 | 4 | *1* | 3 | 11 | 4 | **7** | 15 |
| 10 | *2* | 2 | *3* | **3** | *1* | *3* | **3** | 3 | 4 | *2* | 3 | 5 | *2* | 4 | 6 | *2* | 4 | 10 | *2* | **6** | 14 | **3** | 3 | **22** |

Although layer-0 clearly makes the most use of the memory mechanism, as seen in Table 2 and Table 3, enabling TVLP only on layer-0 does not result in significant performance changes compared to the same-depth baseline.

We observed that, compared to TVLP applied to all layers, layer-0 achieves in this configuration significantly shorter memory lengths, as seen in Table A.1 and Table A.2.

## 5.2 Examples of effective memory of Q in layer-0

Here we present some examples of Q memory lengths in layer-0 from the depth-11 model with TVLP applied to all layers.

**Query**   One of the longer query memory spans at time step 1023 in layer-0 appeared for the following text (shortened with ellipsis), in which the 99% effective memory span of 62 tokens corresponds to a complete sentence and is underlined:

```
...This disorder is caused due to the mutation in the hemoglobin molecule where the
glutamic acid present in the sixth position in a hemoglobin molecule in a beta chain
is replaced by valine.  So, the hemoglobin molecules acquire a change in shape, by
converting into a sickle shape from its normal biconcave shape which reduces the oxygen
carrying capacity of the cells and this shape of molecules cannot be easily passed
through the vascular system and it results in forming a block in small blood vessels
such as capillaries
```

Another example of a longer query memory span at time step 1023 in layer-0 is shown in the text below (shortened with ellipsis), in which the 99% effective memory spans of 56 tokens is underlined:

```
...The director of Hawaii's health department and the registrar of records each has
personally verified that the information on Obama's birth certificate is identical to
that in the state's records, the so-called vault copy.  Given that fact, we are loath
even to engage the fanciful notion that President Obama was born elsewhere, contrary to
the information on his birth certificate, but we note for the record that his mother was
a native of Kansas, whose residents have been citizens of the United States for a very
long time, and whose children are citizens of the United
```

Some examples of the highest effective memories in the last layer (layer-10) can be found in Section A.3.

## 6   Limitations

Although a single random seed was used for the experiments, training through the entire Chinchilla regime is known to be relatively stable and not highly sensitive to random seed variations. Moreover, almost parallel training loss and evaluation BPB trajectories suggest that the observed improvements are not driven by outliers or anomalous runs.

We have seen that while different attention projections, heads, and layers specialize in different memory horizons, many exhibit only very short-term memory. For such heads, the time-varying scan could be further parallelized by approximating it with a data-dependent causal 1D convolution, at the cost of an increase in total algorithmic FLOPs.

Finally, whether the demonstrated FLOP and wall-clock advantage of TVLP translates to larger models remains to be explored.

## 7   Conclusion

We showed that TVLP induces a structured temporal inductive bias that partially substitutes for physical depth in a GPT-3 small-style Transformer, with layers and attention heads segregating into different dynamic memory regimes.

In particular, removing one out of 12 transformer layers and applying TVLP to the remaining ones not only significantly reduces the total number of parameters and the inference memory requirements, but also decreases the evaluation BPB at equal FLOPs and to a lesser extent at equal wall-clock time, as demonstrated on Apple M4 Pro (MPS) and extrapolated to the training speed measured on Nvidia L40S (CUDA).

Exploring the applicability of TVLP to larger models remains an important direction for future work.

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

# A Appendix

## A.1 Training curves

The training loss, the CORE metric and the evaluation BPB extrapolated to Nvidia L40S training speed can be seen in Figure A.1, Figure A.2 and Figure A.3, respectively.

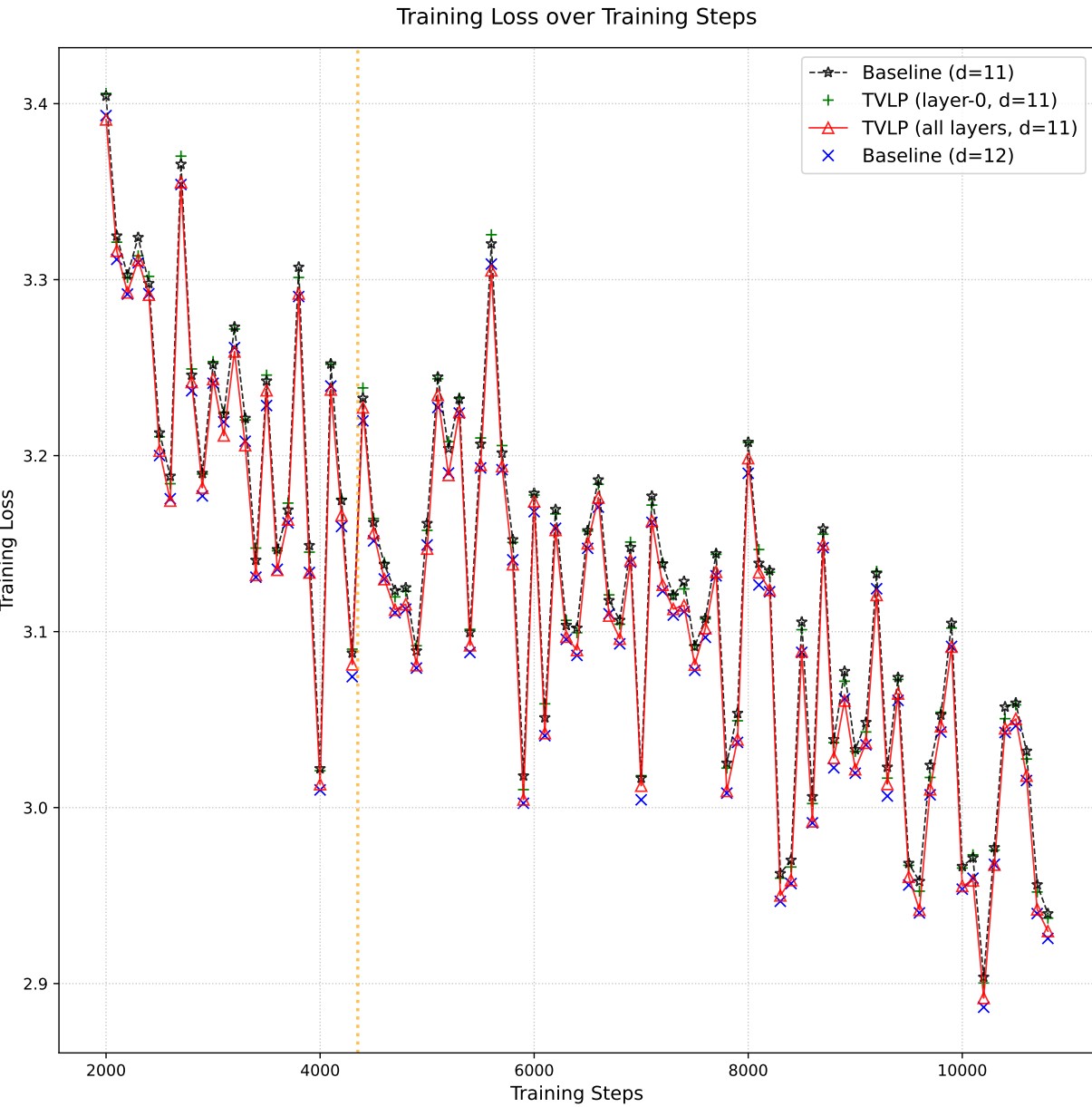

Figure A.1: Training loss of the four model variants of Table 1 trained on Apple M4 Pro (MPS). Loss curves exhibit similar trajectories, suggesting that TVLP provides a systematic improvement rather than altering optimization dynamics. The line styles are like in Figure 1.

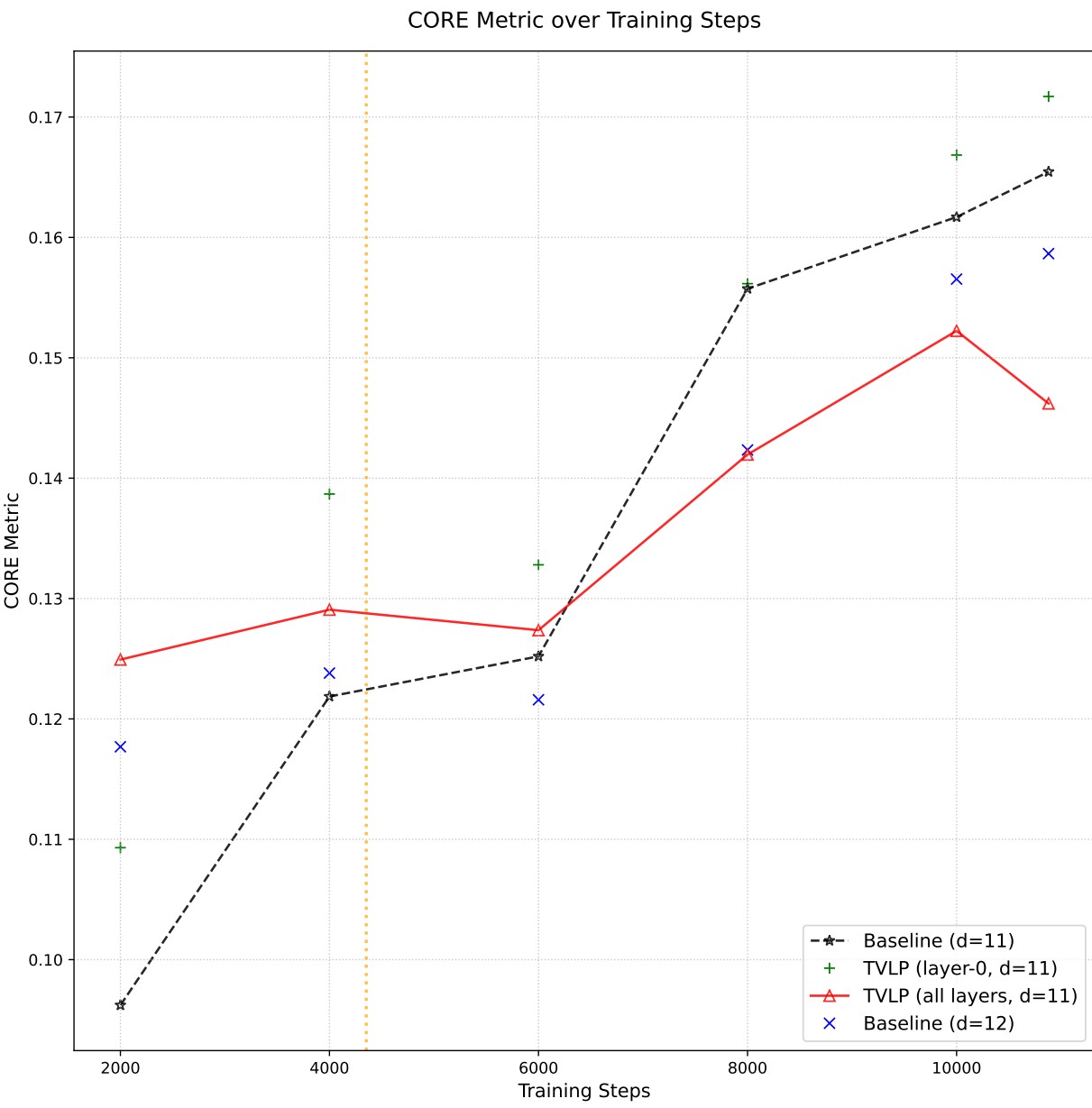

Figure A.2: CORE metric of the four model variants of Table 1 trained on Apple M4 Pro (MPS). The CORE values cover a similar numerical range, within the variance of the metric itself. The line styles are like in Figure 1.

Figure A.3: Evaluation BPB of all four model variants, taken from the training runs on Apple M4 Pro (MPS), but adjusted for the training speed on Nvidia L40S (CUDA) from Table 1. The depth-11 model with TVLP applied to all layers shows lower-or-equal evaluation BPB all throughout, compared to the other models, with a slight advantage at the estimated compute-optimal training time and towards the end of training process. The line styles are like in Figure 1.

### A.2 Effective memory statistics for TVLP applied to layer-0 only

In this subsection we show the effective memory achieved by the depth-11 model with TVLP applied only to layer-0. More exactly, Table A.1 shows the *mean dynamic effective memory length* of layer-0 at time step 1023, while Table A.2 shows the 99-percentile of the *99% effective memory span* at the same time step.

Table A.1: Mean dynamic effective memory length at time step 1023 represented as a vector $\mathbf{m} = (m_Q, m_K, m_V)$ for the depth-11 model with TVLP applied to layer-0. Heads are sorted by increasing vector norm $\|\mathbf{m}\|_2$. Extrema for each of Q, K are underlined and with drawn the font face bold (max) / italic (min).

| Layer | Heads sorted by $\|\mathbf{m}\|_2$ | | | | | | | | | | | | | | | | | | | | | | | |
|---|---|---|---|---|---|---|---|---|---|---|---|---|---|---|---|---|---|---|---|---|---|---|---|---|
| | $m_Q$ | $m_K$ | $m_V$ | $m_Q$ | $m_K$ | $m_V$ | $m_Q$ | $m_K$ | $m_V$ | $m_Q$ | $m_K$ | $m_V$ | $m_Q$ | $m_K$ | $m_V$ | $m_Q$ | $m_K$ | $m_V$ | $m_Q$ | $m_K$ | $m_V$ | $m_Q$ | $m_K$ | $m_V$ |
| 0 | *1.0* | *1.0* | 1.2 | 1.2 | 1.2 | *1.0* | 1.2 | 1.1 | 1.1 | 1.2 | 1.2 | 1.1 | 2.1 | 1.1 | 1.2 | 2.1 | 3.4 | 1.2 | 4.9 | 1.1 | 1.2 | **13.3** | **41.1** | **12.8** |

Table A.2: 99th percentile of the 99% effective memory span at time step 1023, represented as a vector $\mathbf{m} = (m_Q, m_K, m_V)$, for the model with TVLP applied to layer-0. Heads are sorted by increasing vector norm $\|\mathbf{m}\|_2$. Extrema for each of Q, K are underlined and with drawn the font face bold (max) / italic (min).

| Layer | Heads sorted by $\|\mathbf{m}\|_2$ | | | | | | | | | | | | | | | | | | | | | | | |
|---|---|---|---|---|---|---|---|---|---|---|---|---|---|---|---|---|---|---|---|---|---|---|---|---|
| | $m_Q$ | $m_K$ | $m_V$ | $m_Q$ | $m_K$ | $m_V$ | $m_Q$ | $m_K$ | $m_V$ | $m_Q$ | $m_K$ | $m_V$ | $m_Q$ | $m_K$ | $m_V$ | $m_Q$ | $m_K$ | $m_V$ | $m_Q$ | $m_K$ | $m_V$ | $m_Q$ | $m_K$ | $m_V$ |
| 0 | *2* | *2* | 5 | 4 | 4 | *3* | 5 | 4 | *3* | 5 | 4 | *3* | 14 | 3 | 4 | 10 | 17 | 4 | 28 | 3 | 4 | **79** | **225** | **68** |

The achieved memory spans are significantly smaller than those of Table 2 and Table 3.

### A.3 Examples of effective memory span at layer-10

The effective memory span at the last layer is difficult to interpret, in particular due to the possibly additive effect of the effective memory of the previous layers. We focus here on examples of the 99% effective memory span of layer-10, without trying to further interpret how the respective memory is used.

**Queries** One of the longer 99% effective memory horizons for queries at layer-10 at time step 1023 appeared for the following text (shortened with ellipsis), with a 99% effective memory span of 8 tokens:

```
...This article has been published as part of BMC Bioinformatics Volume 10 Supplement
5, 2009: Proceedings of the Bio-Ontologies Special Interest Group Workshop 2008:
Knowledge in Biology. The full contents of the supplement are available online at
http://www.biomedcentral.com/1471-2105/10?issue=S5. The authors declare that they have
no
```

**Keys** One of the higher 99% effective memory lengths for keys at layer-10 at time step 1023 appeared for the following text (shortened with ellipsis), with a 99% effective memory span of 10 tokens:

```
...Me: This is really interesting to me because personally, speech doesn't feel
"synchronous" at all. It feels like I never know when I'm going to
```

**Values** One of the higher 99% effective memory lengths for values at layer-10 at time step 1023 appeared for the following text (shortened with ellipsis), with a 99% effective memory span of 40 tokens:

```
...are also strongly correlated.
|GM|...|0.087 **||0.200 **||0.005||-0.005||0.078 **||-0.006||0.116 **||0.077 **|
|NC|...|0.154 **||0.103 **||0.014||0.011||0.078 **||0.322 **||0.067 **||0.027|
|NPS|...|0.141 **||0.259 **||-0.018||0.028||0.256 **||0.107 **||0.063 **||0.060 **|
```

```
|TPP|...|0.226 **||0.220 **||0.148 **||0.124 **||0.178 **||0.114 **||0.158 **||0.160 **|
|OH|...|0.296 **||0.211 **||0.150 **||0.081 **||0.164 **||0.072 **||0.185 **||0.142 **|
|AP|...|0.471 **||0.211 **||0.296 **||0.188 **||0.167 **||0.114 **||0.238 **||0.197 **|
|CP||0.117 **||0.209 **||0.180 **||0.268 **||0.345 **||0.485 **||1
```

