# OpenReview forum: "Transformer Depth Reduction via Low-Pass Filtering"
_TMLR — Under review for TMLR_

### Review · Reviewer_19pd · 2026-06-28

**Summary Of Contributions:**

The paper introduces Time-Variable Low-Pass filtering (TVLP), an input-dependent exponential moving average applied independently to the query, key, and value projections of each attention head before positional encoding. At each timestep, an input-dependent scalar gate controls the degree of temporal smoothing, allowing individual heads to learn different temporal smoothing horizons. TVLP is implemented using custom sequential Metal kernels and warp-scan CUDA kernels and evaluated using a GPT-3-small-style model. The experiments show that an 11-layer model with TVLP in every layer outperforms the standard 11-layer baseline and nearly matches a 12-layer baseline at equal training steps, while obtaining lower evaluation bits-per-byte at equal estimated FLOPs and slightly better performance at equal estimated wall-clock time. An analysis of the learned filters suggests that heads specialize into different memory regimes, where layer-0 develops heads with long-term memory while deeper layers settle into short-term memory.

### Strengths:
* Simple and computationally inexpensive modification that can be integrated into standard multi-head attention.
* The memory analysis provides useful evidence that different heads and layers learn different temporal smoothing behaviors.
* The wall-clock numbers are measured using efficient custom Metal and CUDA implementations.

### Weaknesses:
* The equal-FLOPs comparison does not independently isolate the benefit of TVLP. Its advantage over the 12-layer baseline follows from using approximately 5.9% fewer FLOPs per token at similar BPB, while the direct gain over the 11-layer baseline is small. All 3 BPB plots are this single result re-plotted, not independent confirmations.
* Each variant is trained with only one seed, despite the key BPB separations being below 0.005. The near-parallel trajectories demonstrate consistency within a run but do not measure the between-seed variance needed to support the claim that TVLP matches the 12-layer baseline at these margins.
* The CORE metric contradicts the BPB results. TVLP applied to all layers is the lowest of the four variants at the final checkpoints. Dismissing this result as noise while treating similarly single-seed BPB curves as robust evidence is not sufficiently justified.
* The cross-hardware claim is overstated because no model was trained to convergence on L40S.
* The structural analysis contradicts the results. Layer-0 has the longest memory yet contributes nothing alone, while the all-layers gain comes from deeper layers whose individual memory is near-trivial.

**Audience:**

Yes

**Audience Explanation:**

A near-free gated EMA on attention projections that recovers roughly one layer's worth of BPB, with a measurable inference-cache reduction, is directly relevant to work on efficient architectures and attention/SSM hybrids. The segregation of heads into distinct memory horizons is of independent interest regardless of whether the depth-substitution claim holds at scale.

**Claims And Evidence:**

No

**Claims Explanation:**

The mechanism, overhead accounting, and cache comparison are well-supported, but the main depth-substitution claims are not. They rest on a single seed per variant with BPB margins under 0.005, the equal-FLOPs and wall-clock axes are one set of runs re-plotted rather than independent confirmations, and the CORE metric points the opposite way at the final checkpoints. The claims exceed what single-seed evidence at these margins can establish.

**Requested Changes:**

### Critical to acceptance
* Clarify that the equal-FLOPs advantage over the 12-layer baseline follows from removing one layer and therefore does not independently demonstrate TVLP’s benefit. The depth-substitution claim ultimately depends on showing that TVLP’s improvement over the same-depth baseline exceeds seed variance.
* Repeat the main 11-layer and 12-layer comparisons across multiple random seeds and report means and uncertainty estimates for BPB at the final and compute-optimal checkpoints.
* Reconcile the CORE result with the BPB conclusion. Analyze why TVLP-all is lowest on CORE at the final checkpoints, or, if CORE is too noisy to interpret at this scale, apply the same noise caveat to the BPB claims and restrict the "effective depth" framing to the observed language-modeling result.
* Either conduct an end-to-end L40S training validation or narrow the multi-hardware wall-clock claims to state explicitly that the L40S result is a throughput-based extrapolation.

### Would strengthen the work
* Measure autoregressive inference latency, generation throughput, and peak memory in addition to theoretical cache size.
* Add layer-subset ablations, such as applying TVLP to contiguous or scattered subsets of deeper layers, to determine which layers produce the gain and connect this result to the effective-memory analysis. Otherwise, Section 5 should be presented as descriptive rather than explanatory.

---

### Review · Reviewer_x9qA · 2026-07-09

**Summary Of Contributions:**

The paper proposes TVLP, an input-dependent exponential moving average applied to the q/k/v projections of each attention head, before RoPE. The gate is a scalar sigmoid of the current token, learned separately per head and per projection. On NanoChat models (about 273M params, Chinchilla regime), a depth-11 model with TVLP roughly tracks a depth-12 baseline in evaluation BPB while clearly beating the depth-11 baseline, at a cost of about 2.4% of one block's parameters. The paper also defines an "effective memory length" statistic and observes that layer-0 heads learn long memories while deeper layers stay short-term. Custom Metal and CUDA kernels are provided.

**Audience:**

Yes

**Audience Explanation:**

People working on efficient attention and gated recurrences would find this interesting even in a narrowly scoped form. The kernels are a nice artifact too.

**Broader Impact Concerns:**

None.

**Claims And Evidence:**

No

**Claims Explanation:**

The parts I could verify directly are fine. I re-derived the overhead numbers (2.42% / 1.61% / 2.11% of one block) from the mechanism and they are exact, and the EMA math in Eqs. 1-6 checks out. My issues are with the headline claims.

(a) Every BPB gap between the four variants is 0.002-0.005, from a single seed, with no error bars. The stability argument in Sec. 6 (near-parallel loss curves) doesn't tell me whether a 0.003 gap survives a rerun. This one issue sits under claims 1, 2 and 5 simultaneously, and it's the main thing between me and a Yes.

(b) The title says "depth reduction" but the evidence is one 12-to-11 comparison at one scale. The width (1024) is also deliberately larger than the compute-optimal 768 the paper itself reports, which should make the 12th layer easier to replace. And "nearly matches" still means worse than the 12L baseline.

(c) On CORE, the only downstream metric (Fig. A.2), TVLP-all-layers is the worst of the four variants at the end of training. The paper waves this off as metric variance but never quantifies it, while citing CORE favorably for the layer-0 variant. I can believe CORE is noisy at 273M, but then the variance needs to be shown.

(d) Sec. 5 is written causally ("substitutes for physical depth") but the evidence is descriptive statistics from the paper's own definitions. Note $\tau$ is essentially $1/\alpha$ in disguise. And TVLP on layer-0 only, the layer with the longest memories, doesn't help at all, which if anything cuts against the story.

**Requested Changes:**

Critical for my recommendation:

1. At least 3 seeds with error bars for all four variants. If the reported gaps hold up, I'll change my Claims And Evidence answer to Yes.
2. Rescope the title and claims to what was shown (one configuration, one scale), or add more depth points, a second scale, or a width-optimal config.
3. Deal with CORE properly: quantify its variance and discuss the ranking inversion, or restrict the claims to BPB.
4. The related work needs a real revision, 12 references is not enough here. The gate has the same form as the forget gate in FoX (Lin et al., ICLR 2025), the update rule is HGRN's (Qin et al., NeurIPS 2023), and EMA-before-attention is MEGA (Ma et al., ICLR 2023) and Megalodon. To be clear, I'm not asking for novelty. But "we introduce" is not accurate as written. The defensible statement is that TVLP makes the EMA-before-attention idea input-dependent and applies it per q/k/v pre-RoPE, and that's fine. Given the title, LayerDrop (Fan et al., ICLR 2020) should also be cited.
5. Either support the causal reading of Sec. 5 (e.g. clamp the layer-0 long-memory head and report the BPB change) or rewrite it as descriptive.

Would strengthen the paper:

6. Some rationale for the design choices: why filter after the projection (vs. the input as in MEGA, or the scores as in FoX), why scalar gates rather than per-channel, why tied gates, why pre-RoPE, why learned initial states. The only ablation is the {Q,K,V} subset sweep at depth 1.
7. The figures can't show a 0.003 gap on a 0.88-0.99 axis. Please plot differences against the 11L baseline or add a zoomed inset.
8. The L40S timing (max over 4 steps, averaged over 8 runs) is a best-case estimator, and TVLP gets hand-tuned kernels while the baseline runs stock. Mean or median over longer warmed runs would be more convincing.
9. Bookkeeping: 10,880 steps at 524,288 tokens over 285M params is about 20 tokens/param, so the "approximately 8" in Sec. 3 looks like a labeling slip. 524,288 tokens/step with B=4 and T=1024 also implies gradient accumulation of 128, never stated. And the Lan et al. reference is missing its title.

---

### Review · Reviewer_dpgp · 2026-07-22

**Summary Of Contributions:**

The paper introduces TVLP (Time-Variable Low-Pass) filtering: an input-dependent causal exponential moving average applied to the Q/K/V projections of attention, before rotary positional encoding. The gating coefficient sets the degree of temporal smoothing per token, and the initial state is a learned parameter. Using the NanoChat harness at GPT-3-small scale, four variants are trained for an identical number of steps. Three main claims are made: (1) At equal steps, depth-11 + TVLP (all layers) attains lower evaluation BPB than the depth-11 baseline and comes close to the depth-12 baseline. (2) At equal FLOPs it outperforms both baselines, and it is never worse in wall-clock terms on Apple M4 Pro (MPS) or at throughputs measured on Nvidia L40S (CUDA). (3) Layer-0 heads segregate into distinct memory regimes, including long-term memory (mean effective memory length 457 steps for K and 489 for V), whereas deeper layers show only short- and medium-term memory. As a secondary contribution the paper implements sequential Metal kernels for MPS and warp-scan CUDA kernels for L40S.

**Audience:**

No

**Audience Explanation:**

## Strengths

- Clear problem framing and honest overhead accounting. The added 304,392 parameters (2.42% of one Transformer block), +0.064 MiB of inference cache, and +0.14% FLOPs/token are measured rather than asserted, which substantiates the "nearly free" premise. The FLOPs model is stated explicitly (3 ops for the forward integration, 4 for the gradients).
- Three separate comparison axes (steps, FLOPs, wall-clock). The paper does not hide the kernel-level overhead, and it explicitly reports the unfavourable fact that TVLP's advantage disappears late in training. This builds credibility.
- Kernel implementations on two distinct backends. The observation that a sequential scan suffices on MPS but lacks parallelism on L40S, and the custom matmul operator written to obtain time-contiguous tensors for the warp scan, constitute a reproducible and practically useful contribution.
- Quantification of effective memory. Presenting both the expectation-based $\tau_t$ (Eq. 5) and the 99% effective memory span (Eq. 6) makes the gate statistics interpretable. The anonymous model source code repository is also appreciated.


## Weaknesses
- A single seed, a single scale, and statistical support that is weak relative to the effect size. The final BPB gap is roughly 0.889 (d11) versus 0.885 (TVLP-all and d12) — about 0.004 BPB. The authors argue that training through the Chinchilla regime is insensitive to seed variation, but no measurement supports this. Without evidence that the gap exceeds seed variance, the central claim does not stand. At least three seeds are needed; if full-length retraining is prohibitive, seed variance for the two baselines and TVLP-all alone would already help.
- No parameter-matched control. TVLP simultaneously removes a layer and adds parameters plus learned initial states. The observed gain therefore cannot be attributed to a temporal inductive bias as opposed to a plain parameter/regularization effect. Needed controls: (i) a depth-11 baseline with slightly larger width or FFN so that parameter counts match; (ii) an EMA with $\alpha$ as an input-independent learned scalar, isolating the contribution of input dependence; (iii) a fixed-$\alpha$ low-pass filter. Control (ii) in particular tests the "time-variable" premise that gives the method its name.
- The structural interpretation contradicts the performance result. Table 2 shows that most heads have mean effective memory lengths of 1.0–1.5, i.e. the filter is nearly the identity almost everywhere; only two heads in layer-0 use substantial memory. Yet the layer-0-only variant performs no better than the baseline (end of Section 5.1). The narrative of Section 5 ("layer-0 forms long-term memory") is thus not causally connected to the gains of Section 4, and the paper itself surfaces this tension while addressing it in a single sentence.
- Missing related work, especially pre-attention EMA. Applying an EMA before attention was proposed as multi-dimensional damped EMA in MEGA (Ma et al., Mega: Moving Average Equipped Gated Attention, ICLR 2023), and input-dependent gating overlaps directly with Mamba's selective mechanism, RG-LRU/Griffin, and gated linear attention. Moreover, the short causal conv1d of Mamba and H3 is essentially the "data-dependent 1D convolution" approximation the authors propose as future work in Section 6. Novelty cannot be assessed without positioning against these; current coverage is limited to a single Mamba citation, RoPE, and Transformer-XL.
- The CORE metric points the other way and is dismissed as variance. In Figure A.2 the depth-11 baseline is consistently above TVLP-all after ~6,000 steps. The claim that this lies "within the variance of the metric itself" is not quantified. Task-level bootstrap confidence intervals are feasible; without them, reporting BPB alone looks selective.
- Internal inconsistency in the training-budget description. Section 3 states that 10,880 steps corresponds to the Chinchilla regime for the depth-12 baseline, but with a token-to-parameter ratio of 8 and 524,288 tokens per step (Figure 2 caption), the compute-optimal point is about 4,350 steps — which is exactly where the dotted line sits in the figures. 10,880 steps is therefore roughly 2.5× Chinchilla (≈20 tokens/parameter). The text should say that training continues well past the Chinchilla optimum.
- Batch configuration is unclear. Section 3 specifies B = 4 and T = 1024, i.e. 4,096 tokens per step, which differs by a factor of 128 from the 524,288 tokens per step used in Figure 2. Gradient accumulation (or a device-vs-global batch distinction) must be stated, since this value sets the scale of every FLOPs axis.
- The L40S result is not a measured training curve. Figure A.3 rescales the MPS curves by the measured L40S throughput ratios. Several places in the paper, including the abstract, say the wall-clock advantage is "confirmed" on L40S; this is extrapolation, not confirmation. The caption is accurate, but the abstract and conclusion should be softened.
- Inference-side cost is not measured. Cache growth is reported, but TVLP adds a sequential state update per decoded token. There is no measurement showing that depth-11 + TVLP actually decodes faster than depth-12. Since inference cost is the main practical motivation for reducing depth, this measurement is close to essential.
- Minor presentation issues. The FLOPs entries in Table 1 use comma decimal separators ("1.371,5×10⁹"), which is easy to misread. Tables 2 and 3 are effectively unreadable at 24 columns; a heatmap or per-layer summary statistics would serve better. The intriguing observation in Section 4 that enabling TVLP on V only caused divergence is left as a single sentence without explanation or follow-up. The rationale for width 1024 over NanoChat's recommended 768 is given, but its effect on the depth-11 vs depth-12 comparison is not discussed.


I do not recommend acceptance in the present form. The mechanism is soundly implemented, the overhead accounting is honest, and the kernel-level contribution is real; but against TMLR's core criterion of claims supported by evidence, three issues remain simultaneously unresolved: (i) an effect size that is small relative to the unquantified seed variance of a single-seed study, (ii) the absence of a parameter-matched control, and (iii) unaddressed relationships to close prior work. If critical changes below are made and the results hold, I would consider this an acceptable paper.

**Broader Impact Concerns:**

None. This is work on training efficiency at small scale, and no separate broader impact statement is required.

**Claims And Evidence:**

No

**Claims Explanation:**

Claims and evidence: only partially supported

The claims about kernel implementation, overhead accounting, and measured throughput are well supported. However, the central claim — that TVLP partially substitutes for physical depth — is not established by the present evidence, primarily for reasons W1–W3 below.

**Requested Changes:**

I would recommend the following:

- Retraining the depth-11 baseline, depth-12 baseline, and TVLP (all layers) with at least three seeds and report mean and standard deviation of final BPB. If full-length runs are infeasible, runs up to the compute-optimal step would be acceptable.
- Adding a parameter-matched depth-11 control and an ablation with α as an input-independent learned parameter. Without the latter, the claim that time-variability drives the gain remains untested.
- Explicitly positioning the method against pre-attention EMA (MEGA) and input-dependent gating (Mamba selective scan, RG-LRU/Griffin, gated linear attention) in a related-work section, with one paragraph stating what is new.
- Correcting the training-budget description and specify the batch configuration in Section 3.
- Softening/toning down the L40S language in the abstract and conclusion to "extrapolated from measured throughput".


The following changes would also strengthen the paper:

- Addressing head-on the mismatch between the structural analysis of Section 5 and the results of Section 4 (the layer-0-only variant yields no gain), ideally with ablations over which layers carry TVLP (e.g. lower half only, upper half only) to localize the source of the improvement.
- Adding task-level bootstrap confidence intervals to the CORE metric.
- Adding measured decoding latency/throughput.
- Adding a second depth pair — e.g. does depth-10 + TVLP match the depth-11 baseline? A single (11, 12) pair is thin support for a general claim about depth substitution.
- Fixing the numeric formatting in Table 1, replace Tables 2–3 with heatmaps, and add a short discussion of the V-only divergence.